# Antibiotic Residues in Animal Products from Some African Countries and Their Possible Impact on Human Health

**DOI:** 10.3390/antibiotics14010090

**Published:** 2025-01-14

**Authors:** Oluwaseun Mary Oladeji, Liziwe Lizbeth Mugivhisa, Joshua Oluwole Olowoyo

**Affiliations:** 1Department of Biology and Environmental Science, Sefako Makgatho Health Sciences University, P.O. Box 139, Pretoria 0204, South Africa; liziwe.mugivhisa@smu.ac.za; 2Department of Health Sciences and The Water School, Florida Gulf Coast University, Fort Myers, FL 33965, USA; jolowoyo@fgcu.edu

**Keywords:** antibiotic, residue, antibiotic resistance, carcinogenicity and policy

## Abstract

This review investigates the levels of antibiotic residues in animal products, types of antibiotics, and their possible impact on human health in Africa. The literature search involved the use of a systematic survey using data that were published from Africa from 2015 to 2024. The search terms used the Boolean operators with keywords such as antibiotics, antibiotic residues, antibiotics in animal products in Africa, and impact on human health. Only research conducted in Africa was used in the present study. The findings showed that the most prevalent groups of antibiotic residues were aminoglycoside, macrolides, β-lactams, fluoroquinolones, tetracyclines sulfonamides, and phenicols. Tetracycline showed the most prevalent antibiotic residue with 43% mostly from East Africa, followed by sulfonamides at 19%, and β-lactams at 16%; most of the antibiotic residue levels were higher than the World Health Organization permissible limit. Noncompliance with withdrawal periods and maximum residue limits for antibiotics used in food-producing animals may lead to negative outcomes such as allergic reactions, teratogenicity, carcinogenicity, microbiome alterations, and, most notably, antibiotic resistance. As a result, there is a need for constant monitoring of antibiotic residues in animal products in addition to the consideration of alternatives to antibiotics in order to avoid their health implications.

## 1. Introduction

Antibiotics are naturally occurring, semi-synthetic, and synthetic agents with antibacterial activity, administered orally, parenterally, or topically [1]. They are employed in both human and veterinary medicine for the prevention and treatment of various diseases, as well as for enhancing growth in cattle [2]. Since their inception in the 1930s, antibiotics have predominantly been utilized to treat or prevent diseases in humans and animals, while also significantly contributing to both food production and contamination through antibiotic residues [3]. The growth-enhancing properties of antibiotics were identified in the 1940s, when animals consuming dried Streptomyces aureofaciens mycelia with chlortetracycline residues exhibited improved growth [4]. In addition to their therapeutic advantages, the 1950 discovery of their ability to promote growth and improve feed efficiency in animals led to their widespread use as feed additives [5]. Antibiotics can accelerate growth by diminishing gut mucous membranes, altering intestinal motility, creating advantageous conditions for good gut microbiota by eradicating pathogenic bacteria, and promoting protein catabolism for muscle growth [6]. They promote growth by reducing immune system activity, hence decreasing nutrient loss and toxic generation [7]. Animals provided antibiotics in their meal often demonstrate a 4–5% rise in body weight compared to animals not receiving drugs [4]. Despite the growth-enhancing properties of antibiotics, they are among the most effective medications for the treatment of different diseases in animals, though their uncontrolled usage may lead to the storage of antibiotics in muscles or other organs of the animals [8,9].

Epidemiological and experimental studies indicate that the effects of antibiotics may accumulate over time in humans, implying that over reliance on antibiotics could foster resistance in the human microbiome [10]. Overreliance on and excessive use of antibiotics for disease treatment may disrupt the delicate equilibrium of beneficial bacteria, thus diminishing diversity and resilience over time. This accumulation may result in a human microbiome that is more susceptible to infections and less capable of responding successfully to new medication [11]. Furthermore, the transmission of these microbiome alterations from parents to their children may set off a cycle in which succeeding generations inherit reduced microbiomes, thereby increasing susceptibility to chronic diseases and antibiotic-resistant infections [12].

One of the greatest threats to public health that faces the whole human population all over the world is the persistent risk of antibiotic contamination [13]. Regardless of national boundaries in terms of geographical location, economy, or legislation, these residues are increasingly becoming a problem. Agricultural industries use a significant amount (50%) of antibiotics for animal production either to treat or prevent possible disease outbreaks in order to maintain animal health [14]. Over the years, several African nations, such as Egypt, Ethiopia, Ghana, Kenya, Nigeria, South Africa, Sudan, and Tanzania, have reported cases of antimicrobial residues [15]. Tetracycline residues were detected in 44% of fresh chicken samples (meat and liver) in Egypt, according to a study that examined the prevalence of these residues (oxytetracycline, tetracycline, chlortetracycline, and doxycycline). The levels of these residues ranged from 38% to 52%. The concentrations of these residues varied from 38% to 52%. There were certain values that exceeded the maximum residue limits set by the Codex, which are 200 μg/kg for chicken meat and 600 μg/kg for liver, respectively. These values were determined by the aggregate of the tetracycline group [16]. The contaminated concentration varied from 103 μg/kg to 8148 μg/kg. When it comes to antibiotic residues found in foods derived from animals, tetracyclines constitute the majority, accounting for 41%, followed by β-lactams, which account for 18% [7]. Considered in light of the fact that these medications are often utilized in the treatment of animals across Africa, this is not surprising. β-Lactams exhibit instability owing to their intrinsic hydrolysis reaction, which may explain their reduced predominance [15]. This paper reviews the prevalence and levels of antibiotic residues in animal-derived food products in Africa from 2015 to 2024 and also highlights the possible health risk. This paper reviews the current trend in antibiotic residues in livestock from African counties. It is the view of the authors that this review will provide needed information to policy makers on the need to have regulations in place guiding the use of antibiotics for livestock production.

## 2. Results and Discussion

### Antibiotic Residues in Animal Products from Africa

Table 1 below shows the concentrations and prevalence of antibiotic residues in animal products in Africa from 2015 to 2024. The results published at the time of compiling this report were only from 10 countries in Africa, comprising 4 countries from West Africa (Nigeria, Cameroon, Ghana, and Benin), 2 from East Africa (Kenya and Ethiopia), 1 from North Africa (Egypt), and 3 from Southern Africa (South Africa, Zambia, and Tanzania). The result presented showed a dearth in information from this region of the world, which may necessitate further research from this region. Previous reports from published work have suggested that research in the field of Environmental Health Science has been limited in this region, which could be attributed to a lack of funding or insufficient information from this region [17]. For instance, most of the reported information on antibiotic residues was from either Nigeria or South Africa. There were about 5 different research studies from Nigeria, 3 from Ethiopia, and 2 from Ghana and South Africa, respectively.

The antibiotic residues detected in milk samples are tetracycline, oxytetracycline, and chlortetracycline, with the concentrations varying from 0.500 μg/kg to 1.569 μg/kg, which were lower than the MRL limit, whereas no samples revealed doxycycline residues [32], as seen in Table 1. From Table 1, it can also be noted that the average concentrations of penicillin G and oxytetracycline residues in the beef were recorded to be 17.58 and 0.24 μg/kg, respectively, according to the study from Cameroon [20]. This concentration is below the MRLs of 50 μg/kg and 600 μg/kg, perhaps due to the low doses of antimicrobials often supplied by pastoralists to optimize the quantity of doses accessible [20]. This study is also similar to one reported from Ethiopia, where penicillin was not discovered and the oxytetracycline levels were lower than MRL [24]; the study conducted in Ethiopia revealed higher oxytetracycline concentrations [23]. The fact that most farmers in the northern region do not follow the dosage instructions for the veterinary medications given to their livestock may help to explain the high percentage of contaminated carcasses [20]. Table 1 further shows that the prevalence of antibiotic residues found in beef (70%) was more than those reported in the northern region of Cameroon (20.30%), according to a study by Mouiche et al. [21]. The overuse of antibiotics, disregard for the time required for withdrawal following antibiotic administration, and disregard for veterinarian advice before antibiotic usage could all contribute to this [19,20]. Currently, there is no definitive legislation regarding the use of antibiotics in animals, and sick animals who do not respond to treatment are frequently murdered for human consumption [40].

Table 1 also shows a study conducted in Ethiopia that aimed to identify specific antibiotic residues in the kidney and red meat of beef cattle butchered at the Nekemte Municipal Abattoir. Oxytetracycline residues were detected in 50% of the samples (15%), whereas tetracycline was present in all 60 (100%) of the analyzed meat samples. Penicillin G, enrofloxacin, sulphadiazine, and doxycycline were not detected in all the samples analyzed in this study. The oxytetracycline residual levels in the kidney and muscle samples ranged from 0.00 to 463.35 µg/kg and 0.00 to 354.55 µg/kg, respectively. In total, 10 percent of the kidney and 3.33 percent of the muscle samples analyzed revealed oxytetracycline residues exceeding the permissible limits. This study’s findings indicated a high level of antimicrobial residue, which suggests that the bacteria may be resistant to antibiotics. This signifies concerted efforts to reduce the impact of the diseases’ resistance on animal and human health [25].

In Nigeria, a report from the study of Anueyiagu et al. [41] indicated that 55.2% (53/96) of the examined animal organs were positive for antibiotic residues, whereas 44.8% (43/96) were negative. The liver exhibited the highest incidence of antibiotic residues at 68.8% (22/32), followed by the kidneys at 56.3% (18/32) and the muscles at 40.6% (13/32) [41], which raised possible concerns for public health. The increased frequency of antibiotic residues in the meat samples may be attributed to cover prescription, overuse, and disregard for the withdrawal time in veterinary practice. On the other hand, antimicrobial medication therapy is thought to be a significant source of antibacterial residue in pigs [42]. Pig farmers’ self-medication practices and the unrestricted availability of antimicrobial medications are the root causes of this issue. Pigs exposed to veterinary medications in an unapproved and careless manner without following recommended dosages are more likely to accumulate violative residues in their tissues. In South Africa, Ramatla et al. [35] investigated antibiotic residues in raw meat using different analytical methods. As shown in Table 1, this study found that the levels of ciprofloxacin and streptomycin residues were higher than the Codex/SA MRL suggested limit, and that the residues of tetracycline, sulfonamide, and streptomycin were positive. Also, Ndlovu et al.’s [36] study demonstrated that the maximum residue levels (MRLs) set by Codex Alimentarius for erythromycin, sulfamethazine, and amoxicillin exceeded the recommended limit. Tetracycline and streptomycin were also identified; however, these levels were below the approved maximum residue limits (MRLs). Approximately 76.6% of samples were over the designated maximum residue limit for sulfamethazine, while 10% exceeded that limit for erythromycin.

Overall, differences in antibiotic residue levels in animal products may be due to animal exposure to antibiotics weeks or days before slaughter, illicit antimicrobial use, excessive drug administration, insufficient farmer knowledge, and/or noncompliance with drug label instructions. This may have to do with the absence of facilities for detection and regulatory agencies that set maximum residue limits (MRLs) to regulate the amount of drug residues in food [43]. This could also be due to the overuse of large amounts of drugs without a professional prescription, the relatively low cost of antibiotics, and the inappropriate administration of antibiotics. Furthermore, this may be due to a lack of awareness and outreach, which could lead to drug addiction and usage, as well as a failure to stick to discontinuation periods [44].

Figure 1 shows the prevalence of antibiotics by group from the available data obtained during this study. From Figure 1, it can be seen that aminoglycoside is the most frequently used antibiotic from the region, followed by macrolides and β-lactams, respectively. From the literature, aminoglycosides are used mostly for the treatment of bacterial infection in livestock. They are often used as a prophylactic to prevent disease outbreaks in livestock, and this might have accounted for the high percentage noted in this study [45]. Apart for its usage as prophylaxis, it has also been used as a growth promoter in some countries, though, recently, a study suggested that this has been largely discouraged owing to the abuse of this drug by the farmers [46]. The study of Ojo et al. [47] from Nigeria showed that 71.4% of poultry farmers in the country used aminoglycosides for farm production, while a study conducted by Kagira et al. [48] previously reported that 61.1% of livestock farmers are using aminoglycosides for livestock production. A similar finding was also reported from South Africa by Kimera et al. [49], that 45 > 5% of farmers rearing cattle used aminoglycosides for their cattle farming. Aminoglycoside residues in meat have been documented to pose serious health risks to consumers, especially in individuals with compromised immune systems [50]. On the other hand, macrolides usage is also common in Africa, and according to Ojo et al. [47], 63.2% of farmers in Nigeria engaging in poultry farming use tylosin and erythromycin, which are classes of macrolides as antibiotics in their poultry production. Apart from the health issue that the residue may pose to human health, macrolides have been reported to accumulate in soil and water and may be bioaccumulated by plants or fish [49,51].

## 3. Impact of Antibiotic Residues on Human Health

Antibiotics administered in animal feed can provide health risks due to their excretion in trace levels into consumable animal tissues. Certain medications can elicit hazardous reactions in consumers directly, while others may induce allergic or hypersensitive responses [52]. β-lactam antibiotics may lead to cutaneous eruptions, dermatitis, gastrointestinal issues, and anaphylaxis, even at low doses. These drugs include the antibiotic classes of penicillin and cephalosporin [53]. The immediate effects may include the emergence of resistance within the normal flora of the human gastrointestinal tract due to the consumption of antibiotic-contaminated meat products, potentially resulting in an outbreak of resistant diarrheal diseases (Figure 2). Furthermore, there is an elevated risk of resistance colonization or infection in humans resulting from their exposure to farm animals administered antibiotics. The indirect and long-term dangers include microbiological effects, cancer potential, reproductive implications, and teratogenic effects [54]. Human health issues that may arise from subchronic exposure levels include allergic reactions in sensitive populations, toxicity, and carcinogenic effects. These effects encompass hypersensitivity reactions, the emergence of resistant strains of bacteria, carcinogenic effects, and possible detrimental impacts on human gut microbiota [55,56]. In terms of human health risks, antibiotic residues may stimulate the spread of antibiotic-resistant bacteria to humans, cause allergies (penicillin), and cause other severe pathologies such as bone marrow toxicity, cancer (furazolidone, oxytetracycline, and sulfamethazine), reproductive disorders (chloramphenicol), mutagenic effects, anaphylactic shock, and nephropathy (gentamicin) [4]. Microbiological impacts are one of the most serious health risks for humans.

Antibiotic-resistant bacteria from animal waste utilized as fertilizer may lead to water supply contamination and modifications in human microbiota [57]. Animal waste frequently harbors microorganisms that have acquired resistance to antibiotics as a result of the extensive application of these medications in animals. When animal dung is sprayed onto crops, resistant bacteria may be transported to the soil and may contaminate local water sources via runoff, particularly after intense rainfall or irrigation. Research indicates that the existence of resistant bacteria in water systems elevates the danger of human exposure via drinking water, recreational activities, and agricultural products [58]. Prolonged exposure to these pollutants can result in heightened antibiotic resistance among human bacterial flora, complicating the treatment of prevalent diseases. Resistant bacteria infiltrating the human gut may supplant beneficial bacteria, potentially resulting in dysbiosis, which has been associated with numerous health complications, including inflammatory bowel illnesses and weakened immunity [59]. Moreover, these bacteria can transfer their resistance genes to other microorganisms in the human microbiome, thereby increasing the incidence of resistant bacteria [60,61].

**Figure 2 antibiotics-14-00090-f002:**
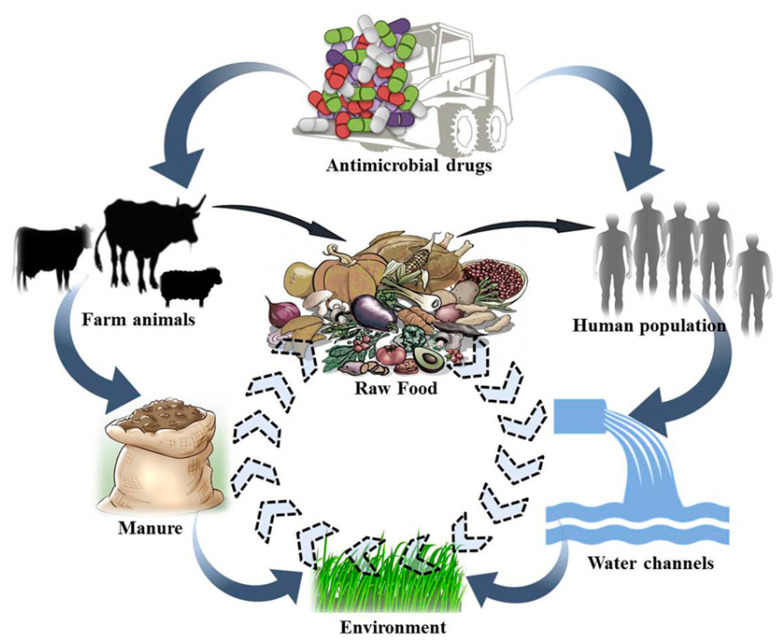
Depiction of potential transfer of antibiotic residues across several ecosystems (adapted from [62]).

### 3.1. Antibiotic Resistance

One of the most serious concerns posed by antibiotic residues is the emergence and spread of antibiotic-resistant microorganisms. When antibiotics are used extensively in agriculture, residues might remain in food products and the environment, allowing bacteria to adapt and acquire resistance [63]. These resistant bacteria possess the capacity to infect humans, resulting in illnesses that are more difficult to manage with conventional remedies. Resistant microorganisms may infiltrate the human body via physical contact or indirectly through eggs, meat, or milk [64,65]. As bacteria of animal origin, they may invade human endogenous flora or add to the reservoir of resistance genes already existing in man. Low dosages of antibiotics have been shown to probably lead to bacterial adaptability, thereby increasing their resistance and severity [66,67]. It is well known that the emergence of antibiotic-resistant bacteria poses a serious danger to human health in many different ways, including hypersensitivity reactions, toxicity, carcinogenicity, and teratogenicity [68,69].

Resistance could be transferred from animals to humans when consuming contaminated meat, milk, or eggs that are not properly cooked or handled, and using manure containing antibiotic residues as fertilizer can spread resistance from animals to humans and back again through soil and water contamination. The use of antibiotics in cattle production has been linked to the development of human antibiotic resistance [57]. The emergence of such resistance poses a direct threat to human health, as pathogens become increasingly resistant to standard antibiotics, complicating treatment options. According to Vieira et al. [70], there is a strong correlation between resistant *Escherichia coli* in food animals (poultry and pigs) and those in human infections, suggesting that many resistant bacteria in human bloodstream infections originate from food animals. Animals treated with low prophylactic antibiotic levels may develop microorganisms resistant to this antibiotic during the production or consumption of animal-derived food. It has been established that humans develop antibiotic-resistant bacteria such as Salmonella, Campylobacter, and Staphylococcus from animal-derived food [57]. These pathogens are commonly transmitted to humans through the consumption of contaminated food, leading to infections that are difficult to treat due to resistance to multiple antibiotic classes [54,71]. As these resistant strains become more prevalent, they present a significant risk to both animal and human health, further emphasizing the need for alternative approaches to antibiotic use in agriculture.

### 3.2. Allergic Reactions

Some individuals may exhibit sensitivity to specific antibiotics and may experience allergic reactions when exposed to antibiotic residues in food. Recent studies have found that antibiotic residues can cause allergic reactions. The bulk of recorded allergies are to beta-lactam antibiotic residues, most notably penicillin and cephalosporins [72]. Erythema multiforme, skin rashes, thrombocytopenia, acute interstitial nephritis, Stevens–Johnson syndrome, vasculitis, hemolytic anemia, serum sickness, and toxic epidermal necrolysis are some of the symptoms that can follow [73]. Allergic reactions have been documented in patients who consumed milk [73], meat [74], and pork, all of which included penicillin residues. Moreover, certain investigations have indicated that residues of aminoglycosides, sulfonamides, and tetracyclines may elicit allergic reactions [4]

### 3.3. Hepatotoxicity Effect

The hepatotoxic effects of certain antibiotics are well documented. Antibiotics like amoxicillin–clavulanate, oxacillin, flucloxacillin, penicillin, and cloxacillin have been linked to cholestatic hepatitis (Hautekeete, [75]). Tetracyclines have been documented to induce hepatotoxicity in animals, especially in livestock and veterinary contexts [76]. Research indicates that elevated doses or the extended administration of tetracyclines in animals, including poultry and cattle, may result in hepatic injury, including cholestasis and fatty liver [77]. Tetracycline may cause acute fatty liver during pregnancy, whereas erythromycin and other macrolides may cause cholestatic hepatitis. Nitrofurantoin has been associated with chronic hepatitis that resembles autoimmune hepatitis [75,78]; studies in laboratory animals have demonstrated that prolonged exposure can cause liver damage akin to that observed in humans, exhibiting signs of cholestatic and hepatocellular injury. Ceftriaxone can generate gallstones, and quinolones and sulfamethoxazole/trimethoprim are linked to significant hepatotoxicity, especially in individuals with AIDS. It is anticipated that large amounts of antibiotic residues may also have detrimental effects on the liver.

### 3.4. Destruction of Normal or Useful Intestinal Flora and Indigestion

The administration of broad-spectrum antibiotics has been linked to the disturbance of the gut microbiota. The intestinal microbiota significantly influences human physiology. They regulate and prevent the colonization of pathogenic microorganisms in the gastrointestinal system. Drugs frequently associated with the diagnosis of gastrointestinal diseases in humans include streptomycin, tylosin, metronidazole, nitroimidazole, and vancomycin [79]. Studies have shown that antimicrobials used for medicinal purposes may modify the ecological composition of gut flora [11]. The gut microbiota comprises numerous bacteria, almost 1000 species, that significantly influence human physiology and health; the bacteria that dwell in the gut function as a barrier, inhibiting germs from colonizing and inducing sickness [80]. Antibiotic residues can kill certain types of bacteria or reduce the number of germs overall. The extent of change, however, is contingent upon the dosage of the antimicrobial agent, method of administration, its bioavailability, metabolic processes, duration of exposure to the drug, and its distribution within the body, including the route of excretion. Children who are exposed to antibiotic residues during their youth experience alterations in the maturation and colonization of their gut microbiota [81], which can result in problems including obesity and an increased risk of allergies [8]. Alzheimer’s disease, inflammatory bowel disease, Anorexia, autism, depression, and Parkinson’s disease are among the neurological conditions that might result from a disruption of microbial homeostasis [82]. Since superbugs cannot be cured, the diseases they cause will be fatal if gut bacteria become resistant to antibiotics and proliferate [83]. Clostridium difficile is also considered to be part of the normal gut microbiota, although the more dominant anaerobes inhibit its growth. Thus, the rate of C. difficile colonization in the human gut varies by age group, with the highest rate occurring in early infancy and decreasing with age [84].

### 3.5. Carcinogenicity and Other Effects

The term carcinogenic denotes any chemical or agent that can modify the genetic structure of an organism, leading to uncontrolled proliferation and malignancy, whereas carcinogen specifically refers to any substance that facilitates carcinogenesis, the process of cancer creation, or exhibiting carcinogenic activity [85]. Antibiotic residues operate by covalently attaching to intracellular components such as DNA, proteins, RNA, phospholipids, glycogen, and glutathione [86]. Sulfamethazine, oxytetracycline, and furazolidone are among the antibiotic residues that can cause cancer [4,7,87]. Apart from their carcinogenic qualities, there have also been reports of other side effects, including nephropathy (mostly associated with gentamicin) and bone marrow toxicity (primarily linked to chloramphenicol) [88]. Beyond these detrimental effects, antibiotic resistance, a secondary consequence of antibiotic residues in food, represents the most severe issue. The WHO projects that, in the absence of intervention, drug-resistant diseases may cause 10 million deaths per year by 2050, leading to economic repercussions akin to those of the 2008–2009 global financial crisis [89].

## 4. Conclusions and Recommendation

This review provides significant insights into the presence and prevalence of antibiotic residues in animal products in Africa and their health implications. The presence of antibiotic residues in food poses a significant public health risk. While crucial in agriculture, their use should be matched with stringent rules to prevent repercussions such as direct toxicity and antibiotic resistance. Comprehensive antibiotic utilization datasets should be established to detect global hotspots of disproportionate antimicrobial usage. Moreover, it is imperative to construct risk assessment methodologies for the prevention of diseases, including the development and dissemination of antimicrobial-resistant bacteria. Similarly, cost-effective and user-friendly technologies must be developed for the quick detection of antibiotic residues in animal products. Ultimately, more beneficial methods like probiotics and herbal treatments ought to be employed. The current study emphasized the need for relevant authorities to take steps to protect consumer health. Controlling antibiotic levels that exceeded maximum residual limits requires a collaborative and coordinated effort by government officials, veterinarians, and animal producers. The animal farm owner, manager, or herder must work with the farm veterinarian to develop treatment procedures that assure the appropriate use of antibiotics. When antibiotics are administered, protocols must be set to teach workers on how to handle the animal safely to avoid unintentional meat residues. Animals undergoing treatment must be recognized, and antibiotic administration must be documented to prevent residues. This indicates the need for relevant authorities to take steps to protect consumer health. Controlling antibiotic levels that exceeded maximum residual limits requires a collaborative and coordinated effort by government officials, veterinarians, and animal producers. The animal farm owner, manager, or herder must work with the farm veterinarian to develop treatment procedures that assure the appropriate use of antibiotics. When antibiotics are administered, protocols must be set to teach workers about how to handle the animal safely in order to avoid unintentional meat residues.

## 5. Limitations of This Study

This review does not have detailed information on the levels of antibiotic residues in different parts of Africa, which makes it harder to find high-risk areas. It also does not give specific plans or steps for putting the suggested risk assessment methods and detection tools into action.

## 6. Materials and Methods

This review adopted a systematic literature review approach using published data from research studies focused on antibiotics in livestock products in Africa from 2015 to 2024. Relevant studies published from 2015 to 2024 that were focused on antibiotics only and mainly conducted in Africa were included in this review. For the literature search, we used Boolean operators such as OR, AND, and NOT as the main tool, and the search involved using key search terms such as antibiotics, antibiotic residues, antibiotics in animal products in Africa, and impact on human health. The search was conducted in major databases such as Google Scholar, ScienceDirect, ResearchGate, Scopus, and PubMed. Each database was searched separately, and the results were compiled and compared for comprehensiveness. For the initial survey, a total of 335 searches were returned, which was further screened down to 167, as seen in Figure 3, following the eligibility criteria listed below:


**
*Eligibility criteria*
**


Articles that met the following criteria were eligible for this review:
Those where the antibiotics analyzed were specified;Those where the concentrations of antibiotic residues in animal products in Africa were quantified;The impact of the antibiotic residues on human health;Research activities mainly carried out in Africa;Published papers that dealt with antibiotics in livestock only.


**
*Data extraction*
**


Table 2 displays the maximum antibiotic residue limit. For ease of comparison, the concentrations of antibiotic residues were converted to µg/kg and summarized, along with the following details: the country in which the study was conducted, the organs used, the first author, and the year of publication. For convenience, countries were arranged in alphabetical order in the table.

## Figures and Tables

**Figure 1 antibiotics-14-00090-f001:**
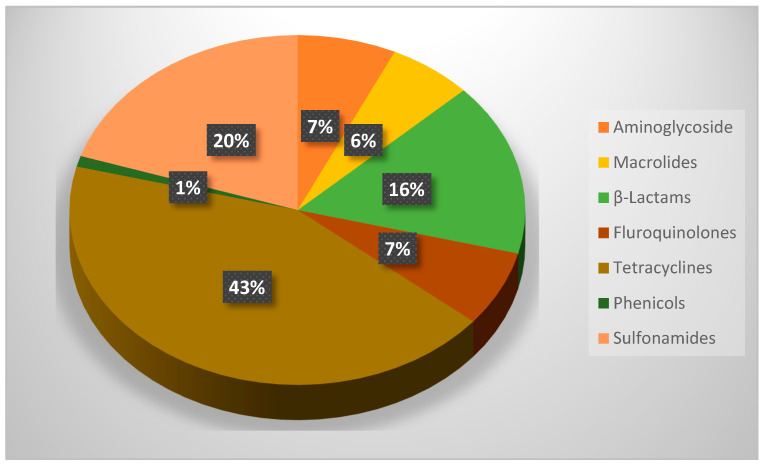
Showing the prevalence of antibiotic residues in this study.

**Figure 3 antibiotics-14-00090-f003:**
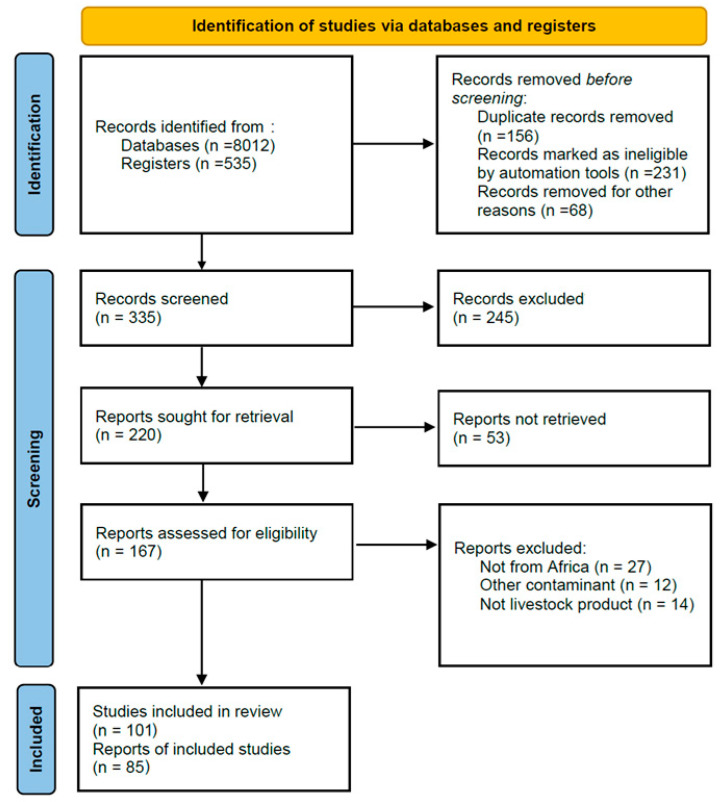
Meta-analysis flowchart for the eligibility of the information source.

**Table 1 antibiotics-14-00090-t001:** Prevalence of antibiotic residues in animal products consumed in some African countries.

**Country**	**Region**	**Source**	**Animal Product**	**Antibiotics**	**Concentration (µg/kg)**	**References**
Benin	West Africa	Fish	Muscles	Amphenicols, Macrolids Tetracycline	Not detected Detected	[18]
Cameroun	Middle Africa	Chicken	Liver Muscle	Tetracycline	150 ± 30 62.4 ± 15.3	[19]
Cameroun	Middle Africa	Cow	Beef	Penicillin Oxytetracycline	17.580 240	[20]
Cameroon	Middle Africa		Raw egg, raw milk and beef	Antibiotic present	Detected	[21]
Egypt		Cow	Raw beef	Gentamicin β-lactam Ciprofloxacin Erythromycin Oxytetracyclines Sulfonamide	46–112.7 20–67.73 56.46–81.91 34.29–45.54 19.9–119 12.67–19.33	[22]
Egypt	Northern Africa	Cow	Beef	Aminoglycoside Penicillin G Ciprofloxacin Macrolides Oxytetracycline Sulfonamide	46–76 20–55 56.45–81.91 29.29–41.38 19.9–86.7 12.67–19.33	[22]
Ethiopia	East Africa	Cattle	Kidney Muscle	Tetracycline	16–43 6–43	[23]
Ethiopia	East Africa	Cow	Beef	Tetracycline Penicillin G Oxyteracycline	9.35 Not Detected 15.99 ± 13.2–22.32 ± 2.38	[24]
Ethiopia	East Africa		Kidney Muscle	Oxytetracycline Doxycycline, Sulphadiazine, penicillin G, and Enrofloxacin	0–463.35 0–354.55	[25]
Ghana	West Africa	Cow, Chicken	Beef Chicken Egg	Tetracycline, Oxytetracycline, Chlortetracycline, Amoxicillin, Cefazolin, penicillin G), sulfamethoxazole, sulfadoxine, sulfathiazole	81.35 76.94 234.43 35.76 47.02 41.02 103.98 46.05 68.63	[26]
Kenya	East Africa		Milk	β-lactam	Detected	[27]
Kenya	East Africa		Milk	Amoxicillin Cloxacillin Tetracycline Sulfamethoxazole Trimethoprim	6.7 53.3 30.6 5.0 6.2	[28]
Kenya	East Africa	Chicken	Meat	Sulfonamide Sulfadiazine Sulfamethazine	39.91–101.39 35.21 39.34	[29]
Nigeria	West Africa		Fresh Milk Cheese Fermented Milk	Penicillin G	15.22 ± 0.61 7.60 ± 0.60 8.24 ± 0.50	[30]
Nigeria	West Africa		Milk	Tetracycline	Detected	[31]
Nigeria	West Africa		Milk	Tetracycline, oxytetracycline chlortetracycline Doxycycline	0.01–1.57 Not detected	[32]
Nigeria	West Africa	Cattle	Meat Liver Kidney	Oxytetracycline	Above MRL	[33]
Nigeria	West Africa	Cattle	Beef	Tetracycline Ciprofloxacin Oxytetracycline	17.57 ± 6.20–594 ± 47.71 22.4 ± 5.20–82.77 ± 12.60 227.2 ± 16.45–474.4 ± 119.74	[34]
South Africa	Southern Africa	Chicken Cow Pig	Chicken Muscle Liver Beef Muscle Liver Kidney Muscle Kidney Liver Muscle Liver Muscle Liver Kidney Muscle Liver Kidney Muscle Liver Muscle Liver Kidney Muscle Liver Kidney Muscle Liver Muscle Liver Kidney Muscle Liver Kidney	Ciprofloxacin Streptomycin Sulfonamide Tetracycline	89.6–175.9 152.2–289.1 89.6–146.1 145.2–316.5 98.2–197.0 42.6–95.8 72.5–140.2 220.0–355.6 98.4–452.9 368.8–986.4 625.9–989.2 498.2–920.1 614.2–1280.6 620.3–875.8 196.5–535.9 14.2–1052.6 32.5–65.9 45.8–81.6 – 19.8–87.9 37.6–73.9 – 48.2–69.9 52.8–92.8 41.2–82.1 42.56–286.2 26.6–61.5 41.2–221.6 41.2–359.2 46.67–86.9 101.3–489.1 43.7–255.9	[35]
South Africa	Southern Africa	Goat	Milk	Tetracycline and Streptomycin	Below MRLs	[36]
Tanzania	East Africa	Chicken	Muscle Liver Kidney	Tetracycline	2604.1 ± 703.7 3434.4 ± 604.4 3533.1 ± 803.6	[37]
Tanzania	East Africa	Cow	Beef	Oxytetracycline	0.69 ± 0.09	[38]
Zambia	East Africa	Cow	Beef	Oxytetracycline Sulfamethazine	27.26–481.61 11.92–259.98	[39]

**Table 2 antibiotics-14-00090-t002:** Maximum allowable residue limits of antibiotics in products of animal origin [90].

**Antibiotics**	**Animal Species**	**Tissues**	**MRL (** **μg/kg)**
Ampicillin	All food-producing species	Milk Liver Fat Kidney Muscle	4 50 50 50 50
Amoxicillin	All food-producing species	Milk Liver Fat Kidney Milk	4 50 50 50 50
Avilamycin	Poultry, porcine, rabbit	Kidney Liver Fat Muscle	200 300 100 50
Benzylpenicillin	All food-producing species	Milk Liver Fat Kidney Muscle	4 50 50 50 50
Bacitracin	Bovine	Milk	100
Cefapirin	Bovine	Muscle Fat Kidney Milk	50 50 100 60
Clavulanic acid	Bovine, porcine	Kidney Fat Liver Muscle	400 100 200 100
Cefacetrile	Bovine	Milk	125
Cefazolin	Bovine, ovine, caprine	Milk	50
Cloxacillin	All food-producing species	Milk Liver Fat Kidney Muscle	30 300 300 300 300
Chlortetracycline	All food-producingspecies	Muscle Liver Kidney Milk Eggs	100 300 600 100 200
Clavulanic acid	Bovine, porcine	Muscle Fat Liver Kidney	100 100 200 400
Cloxacillin	All food-producing species	Muscle Fat Liver Kidney Milk	300 300 300 300 30
Colistin	All food-producing species	Muscle Fat Liver Kidney Milk Eggs	150 150 150 200 50 300
Doxycycline	Bovine Porcine, poultry	Muscle Liver Kidney	100 300 600
		Not for use in animals from which milk is produced for human consumption	
		Muscle Skin and fat Liver Kidney	100 300 300 600
		Not for use in animals from which eggs are produced for human consumption	
Dicloxacillin	All food-producing species	Muscle Fat Liver Kidney Milk	300 300 300 300 30
Erythromycin A	All other food-producing species	Muscle Fat Liver Kidney Milk Eggs	200 200 200 200 40 150
Gentamicin	Bovine, porcine	Muscle Fat Liver Kidney Milk	50 50 200 750 100
Kanamycin A	All food-producing species except fin fish	Muscle Fat Liver Kidney Milk	100 100 600 2500 150
Oxytetracycline	All food-producing species	Muscle Liver Kidney Milk Eggs	100 300 600 100 200
Oxacillin	All food-producing species	Muscle Fat Liver Kidney Milk	300 300 300 300 30
Sulfonamides	All food-producing species	Muscle Fat Liver Kidney	100 100 100 100
Streptomycin	All ruminants, porcine, rabbit	Muscle Fat Liver Kidney	500 500 500 1000
Tylosin A	All food-producing species	Muscle Fat Liver Kidney Milk Egg	100 100 100 100 50 200

## Data Availability

All data are provided in the manuscript.

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
