# Peer review of "Antibiotic Residues in Animal Products from Some African Countries and Their Possible Impact on Human Health"

_antibiotics, 2025, doi:10.3390/antibiotics14010090_

Round 1
Reviewer 1 Report
Comments and Suggestions for Authors
The manuscript entitled "Antimicrobial residues in animal products from Africa and its impact on human health" is written as per journal style. However, minor corrections and updates are required.
[1]. Please update all the references. for example please check the reference collected using keywords
1: Kilusungu ZH, Kassam D, Kimera ZI, Mgaya FX, Nandolo W, Kunambi PP, Ulomi W, Matee MIN. Tetracycline and sulphonamide residues in farmed fish in Dar es Salaam, Tanzania and human health risk implications. Food Addit Contam Part B Surveill. 2024 Jun;17(2):161-170. doi: 10.1080/19393210.2024.2331106. Epub 2024, Mar 22. PMID: 38516743.
2: Alhaji NB, Odetokun IA, Jibrin MS, Lawan MK, Kwaga J. Antibiotic resistance and mitigation using One Health lens in aquaculture of Northern Nigeria.
Onderstepoort J Vet Res. 2024 Oct 16;91(2):e1-e11. doi: 10.4102/ojvr.v91i2.2165. PMID: 39494642; PMCID: PMC11538107.
3. Hambolu DA, Olatoye OI, Besong MA, Call DR. Low-cost biosecurity measures are associated with reduced detection of non-Typhoidal Salmonella in Nigerian poultry while inappropriate antibiotic use is widespread. Sci Rep. 2024 Sep 9;14(1):20974. doi: 10.1038/s41598-024-72317-y. PMID: 39251698; PMCID: PMC11385543.
4: Moffo F, Ndebé MMF, Tangu MN, Noumedem RNG, Awah-Ndukum J, Mouiche MMM. Antimicrobial use, residues and resistance in fish production in Africa:
systematic review and meta-analysis. BMC Vet Res. 2024 Jul 10;20(1):307. doi:10.1186/s12917-024-04158-w. PMID: 38987775; PMCID: PMC11234786.
5: Smith LC, Stringer A, Owuor KO, Ndenga BA, Winter C, Gerken KN. Moving milk and shifting risk: A mixed methods assessment of food safety risks along
informal dairy value chains in Kisumu, Kenya. One Health. 2024 Oct 13;19:100914. doi: 10.1016/j.onehlt.2024.100914. PMID: 39507304; PMCID: PMC11539344.
6: Tufa TB, Amenu K, Fasil N, Regassa F, Beyene TJ, Revie CW, Hogeveen H,Stegeman JA. Prudent use and antimicrobial prescription practices in Ethiopian
veterinary clinics located in different agroecological areas. BMC Vet Res. 2024, Nov 29;20(1):538. doi: 10.1186/s12917-024-04380-6. PMID: 39614253; PMCID:
PMC11605951.
7: Chebii F, K'oreje K, Okoth M, Lutta S, Masime P, Demeestere K. Occurrence and environmental risks of contaminants of emerging concern across the River Athi Basin, Kenya, in dry and wet seasons. Sci Total Environ. 2024 Mar 1;914:169696. doi: 10.1016/j.scitotenv.2023.169696. Epub 2023 Dec 29. PMID: 38160815.
[2]. Check the table 1 and 2, use the data as mean ± SD. Ensure consistency in units (e.g., μg/kg) within and the number should be consistent (after decimal point) with units. Please do not mix μg/kg with Percent data or other.
[3]. Reformat Section 2.1 and other sections: Check the formatting as per Journal style (reformat the paragraphs).
4. Update the Table and Text related to Ciprofloxacin with the following references:
1: González N, Abdellati S, De Baetselier I, Laumen JGE, Van Dijck C, Block T, Manoharan-Basil SS, Kenyon C. Ciprofloxacin Concentrations 1/1000th the MIC Can Select for Antimicrobial Resistance in <i>N. gonorrhoeae</i>-Important implications for Maximum Residue Limits in Food. Antibiotics (Basel). 2022 Oct
18;11(10):1430. doi: 10.3390/antibiotics11101430. PMID: 36290088; PMCID: PMC9598464.
2: Mdegela RH, Mwakapeje ER, Rubegwa B, Gebeyehu DT, Niyigena S, Msambichaka V, Nonga HE, Antoine-Moussiaux N, Fasina FO. Antimicrobial Use, Residues,nResistance and Governance in the Food and Agriculture Sectors, Tanzania. Antibiotics (Basel). 2021 Apr 16;10(4):454. doi: 10.3390/antibiotics10040454. PMID: 33923689; PMCID: PMC8073917.
3: Moffo F, Mouiche MMM, Djomgang HK, Tombe P, Wade A, Kochivi FL, Dongmo JB, Mbah CK, Mapiefou NP, Mingoas JK, Awah-Ndukum J. Associations between antimicrobial use and antimicrobial resistance of Escherichia coli isolated from poultry litter under field conditions in Cameroon. Prev Vet Med. 2022
Jul;204:105668. doi: 10.1016/j.prevetmed.2022.105668. Epub 2022 May 14. PMID: 35613518.
4: Dahshan H, Abd-Elall AM, Megahed AM, Abd-El-Kader MA, Nabawy EE. Veterinary antibiotic resistance, residues, and ecological risks in environmental samples obtained from poultry farms, Egypt. Environ Monit Assess. 2015 Feb;187(2):2. doi: 10.1007/s10661-014-4218-3. Epub 2015 Jan 20. PMID: 25600402.
[5]. Update the title of manuscript as sulfonamides and fluoroquinolones are antimicrobial not antibiotics.
Author Response
Thank you for your valuable comment.
Reviewer 1
Comments and Suggestions for Authors
The manuscript entitled "Antimicrobial residues in animal products from Africa and its impact on human health" is written as per journal style. However, minor corrections and updates are required.
[1]. Please update all the references. for example please check the reference collected using keywords
1: Kilusungu ZH, Kassam D, Kimera ZI, Mgaya FX, Nandolo W, Kunambi PP, Ulomi W, Matee MIN. Tetracycline and sulphonamide residues in farmed fish in Dar es Salaam, Tanzania and human health risk implications. Food Addit Contam Part B Surveill. 2024 Jun;17(2):161-170. doi: 10.1080/19393210.2024.2331106. Epub 2024, Mar 22. PMID: 38516743.
2: Alhaji NB, Odetokun IA, Jibrin MS, Lawan MK, Kwaga J. Antibiotic resistance and mitigation using One Health lens in aquaculture of Northern Nigeria.
Onderstepoort J Vet Res. 2024 Oct 16;91(2):e1-e11. doi: 10.4102/ojvr.v91i2.2165. PMID: 39494642; PMCID: PMC11538107.
3. Hambolu DA, Olatoye OI, Besong MA, Call DR. Low-cost biosecurity measures are associated with reduced detection of non-Typhoidal Salmonella in Nigerian poultry while inappropriate antibiotic use is widespread. Sci Rep. 2024 Sep 9;14(1):20974. doi: 10.1038/s41598-024-72317-y. PMID: 39251698; PMCID: PMC11385543.
4: Moffo F, Ndebé MMF, Tangu MN, Noumedem RNG, Awah-Ndukum J, Mouiche MMM. Antimicrobial use, residues and resistance in fish production in Africa:
systematic review and meta-analysis. BMC Vet Res. 2024 Jul 10;20(1):307. doi:10.1186/s12917-024-04158-w. PMID: 38987775; PMCID: PMC11234786.
5: Smith LC, Stringer A, Owuor KO, Ndenga BA, Winter C, Gerken KN. Moving milk and shifting risk: A mixed methods assessment of food safety risks along
informal dairy value chains in Kisumu, Kenya. One Health. 2024 Oct 13;19:100914. doi: 10.1016/j.onehlt.2024.100914. PMID: 39507304; PMCID: PMC11539344.
6: Tufa TB, Amenu K, Fasil N, Regassa F, Beyene TJ, Revie CW, Hogeveen H,Stegeman JA. Prudent use and antimicrobial prescription practices in Ethiopian
veterinary clinics located in different agroecological areas. BMC Vet Res. 2024, Nov 29;20(1):538. doi: 10.1186/s12917-024-04380-6. PMID: 39614253; PMCID:
PMC11605951.
7: Chebii F, K'oreje K, Okoth M, Lutta S, Masime P, Demeestere K. Occurrence and environmental risks of contaminants of emerging concern across the River Athi Basin, Kenya, in dry and wet seasons. Sci Total Environ. 2024 Mar 1;914:169696. doi: 10.1016/j.scitotenv.2023.169696. Epub 2023 Dec 29. PMID: 38160815.
Response: Thank you for the valuable comment. After carefully checking the references advices, they did not meet the Eligibility criteria used for the review stated in material and method of these manuscript and the study focus on Antibiotic residues not Antimicrobial residues.
[2]. Check the table 1 and 2, use the data as mean ± SD. Ensure consistency in units (e.g., μg/kg) within and the number should be consistent (after decimal point) with units. Please do not mix μg/kg with Percent data or other.
Authors Response: The table has been carefully checked.
[3]. Reformat Section 2.1 and other sections: Check the formatting as per Journal style (reformat the paragraphs).
Authors Response: Section 2.1 and other section has been formatted to Journal style.
- Update the Table and Text related to Ciprofloxacin with the following references:
1: González N, Abdellati S, De Baetselier I, Laumen JGE, Van Dijck C, Block T, Manoharan-Basil SS, Kenyon C. Ciprofloxacin Concentrations 1/1000th the MIC Can Select for Antimicrobial Resistance in <i>N. gonorrhoeae</i>-Important implications for Maximum Residue Limits in Food. Antibiotics (Basel). 2022 Oct
18;11(10):1430. doi: 10.3390/antibiotics11101430. PMID: 36290088; PMCID: PMC9598464.
2: Mdegela RH, Mwakapeje ER, Rubegwa B, Gebeyehu DT, Niyigena S, Msambichaka V, Nonga HE, Antoine-Moussiaux N, Fasina FO. Antimicrobial Use, Residues,nResistance and Governance in the Food and Agriculture Sectors, Tanzania. Antibiotics (Basel). 2021 Apr 16;10(4):454. doi: 10.3390/antibiotics10040454. PMID: 33923689; PMCID: PMC8073917.
3: Moffo F, Mouiche MMM, Djomgang HK, Tombe P, Wade A, Kochivi FL, Dongmo JB, Mbah CK, Mapiefou NP, Mingoas JK, Awah-Ndukum J. Associations between antimicrobial use and antimicrobial resistance of Escherichia coli isolated from poultry litter under field conditions in Cameroon. Prev Vet Med. 2022
Jul;204:105668. doi: 10.1016/j.prevetmed.2022.105668. Epub 2022 May 14. PMID: 35613518.
4: Dahshan H, Abd-Elall AM, Megahed AM, Abd-El-Kader MA, Nabawy EE. Veterinary antibiotic resistance, residues, and ecological risks in environmental samples obtained from poultry farms, Egypt. Environ Monit Assess. 2015 Feb;187(2):2. doi: 10.1007/s10661-014-4218-3. Epub 2015 Jan 20. PMID: 25600402.
Authors Response: Thank you for the comment. After carefully checking the references advices, they did not meet the Eligibility criteria used for the review stated in material and method in the manuscript under Ciprofloxacin.
[5]. Update the title of manuscript as sulfonamides and fluoroquinolones are antimicrobial not antibiotics.
Response: Thank you for the comment. In this present reviews the highest antibiotics reviews were tetracycline with 43%. The findings showed that the most prevalent group of antibiotic residues were aminoglycoside, macrolides, β-lactams, fluoroquinolones, tetracyclines sulfonamides, and phenicols.
The title has been modify and now reads “Antibiotics residues in animal products from some Africa countries and it’s possible impact on human health.
S
Reviewer 2 Report
Comments and Suggestions for Authors
Dear Authors and Editor
The authors in this manuscript have scientific merit that warrants publication in Antibiotics (Antibiotic residues in animal products from Africa and its impact on human health). The authors have pointed out firstly the explores antibiotic residues in animal products in Africa and covers multiple African regions their impact on human health, providing valuable insights into a critical public health issue. Overall, the study undertaken by the authors is relevant and significant to the impact of antibiotics to health and recommending how to deal of these and control in these variant reproductive animal farms. These will help the health sectors and veterinarian in optimizing guidelines for using antibiotics.
However, the manuscript needs to update the title such as instead of from Africa to some countries in Africa and also need to add Flow chart Figure of (Meta-Analysis Flow Chart) in the of material and methods and also add the limitation of works. In addition, some sentences need to rewritten the lines 72-77, 102-104, 130-136.
Author Response
Dear Reviewer,
Thank you for your valuable comments.

Reviewer 3 Report
Comments and Suggestions for Authors
Title: Antibiotic residues in animal product from Africa and its impact on human health.
Thanks to the authors for presenting this important subject.
I have a few comments.
Can authors kindly change the word antibiotics to antimicrobials in the title and subsequent parts in the manuscript.
Introduction:
Line 52: antibiotics may be cumulative in humans.
This statement needs to be reworded.
Line 68: reported case of antimicrobials residues.
Was the antimicrobial residues reported above the accepted limit or below?
Lines 186-225: The impact of antibiotic residues on human health
Please remember to consider easy access to antimicrobials for human use first before looking at the veterinary use. This area of research is not that straightforward to understand.
Lines 226-255: Antibiotic resistance
Please remember that antimicrobial residues/resistance can be transmitted from animals to humans and vice-versa. Please reconsider in all the sections of your manuscript. It is not always easy to show which the direction of infection is coming from.
Lines 282-302. Destruction of normal intestinal flora and indigestion. Please consider practical diseases such as Clostridium difficile infection in both humans and Livestock.
Author Response
Dear Reviewer,
Thank you for your valuable comments.
--------------------------------------------------
Reviewer 3
I have a few comments.
Can authors kindly change the word antibiotics to antimicrobials in the title and subsequent parts in the manuscript.
Author Response: Thank you for your valuable comment. The study primarily examines the use of antibiotics in animal agriculture in Africa, which may lead to antibiotic residues in food, posing a risk to public health.
Introduction:
Line 52: antibiotics may be cumulative in humans.
This statement needs to be reworded.
Authors Response: Thanks for the comment the statement has been change to “antibiotics may accumulate over time in humans”.
Line 68: reported case of antimicrobials residues.
Was the antimicrobial residues reported above the accepted limit or below?
Authors Response: Thank you for your comment, most of the antibiotic residue levels were higher than maximum residue limits set by Codex as seen in line 72 and also reported in the discussion section.
Lines 186-225: The impact of antibiotic residues on human health
Please remember to consider easy access to antimicrobials for human use first before looking at the veterinary use. This area of research is not that straightforward to understand.
Author Response: Thank you for the comment, the reason why we start with veterinary is that, these residues typically come from the use of antibiotics in agriculture, aquaculture, and veterinary medicine and there is an elevated risk of resistance colonization or infection in humans resulting from their exposure to farm animals administered antibiotics.
Lines 226-255: Antibiotic resistance
Please remember that antimicrobial residues/resistance can be transmitted from animals to humans and vice-versa. Please reconsider in all the sections of your manuscript. It is not always easy to show which the direction of infection is coming from.
Authors Response: Thank you for the comment, yes antimicrobial residues/resistance can be transmitted from animals to humans and vice-versa, it included in the manuscript.
Lines 282-302. Destruction of normal intestinal flora and indigestion. Please consider practical diseases such as Clostridium difficile infection in both humans and Livestock.
Authors Response: Thank you for the valuable comment. The clostridium difficile disease has also been included in the manuscript. “Clostridium. difficile is also considered as part of the normal gut microbiota, although the more dominant anaerobes inhibit its growth. Thus, the rate of C. difficile colonization in the human gut varies by age group, with the highest rate occurring in early infancy and decreasing with age [84].